# Adaptive Kalman Filtering for Compensating External Effects in On-Line Spectroscopic Measurements

**DOI:** 10.3390/s25082513

**Published:** 2025-04-16

**Authors:** Daniel Sbarbaro, Tor Arne Johansen, Jorge Yañez

**Affiliations:** 1Department of Electrical Engineering, Faculty of Engineering, Universidad de Concepcion, Concepción 4070371, Chile; 2Center for Autonomous Marine Operations and Systems (AMOS), Department of Engineering Cybernetics, Norwegian University of Science and Technology, 7491 Trondheim, Norway; tor.arne.johansen@ntnu.no; 3Department of Analytical and Inorganic Chemistry, Faculty of Chemical Sciences, Universidad de Concepcion, Concepción 4070371, Chile; jyanez@udec.cl

**Keywords:** spectroscopy, Kalman filter, sensor calibration

## Abstract

This study addresses the challenges of real-time spectroscopic sensing in industrial applications, where external factors such as temperature fluctuations, pressure variations, and particle size distribution significantly impact measurement accuracy. Conventional quantitative analytical methods often neglect these dynamic influences, leading to erroneous concentration estimates. To overcome these limitations, we propose an integrated modeling framework that combines a discrete-time process model with a physics-based spectroscopic sensor model, explicitly accounting for the dynamic properties of the system. A key innovation of this work is the development and application of an Adaptive Kalman Filter (AKF) to systematically correct for measurement distortions caused by external disturbances. Unlike conventional filtering techniques, the AKF dynamically adjusts to changing process conditions by leveraging real-time observability analysis, ensuring robustness even in the presence of sensor noise and environmental variability. Furthermore, to address cases where full observability is not achievable, we introduce a reduced-order Adaptive Kalman Filter (rAKF), which optimally estimates concentrations while minimizing computational complexity. A comprehensive series of simulations was conducted to assess the sensitivity of the estimation to variations in external signal type, noise levels, and initial values for parameters and states. The findings of this study demonstrate the superior performance of both AKF and rAKF in comparison to conventional filtering techniques, including the Extended Kalman Filter. The proposed approaches have been shown to enhance the reliability of spectroscopic sensor measurements, enabling more precise real-time estimations that can be used for monitoring and advanced process control strategies in industrial settings.

## 1. Introduction

Spectroscopic sensors, such as infrared and Raman spectroscopy-based sensors, play a crucial role in both laboratory research and industrial applications, where they are extensively employed for chemical analysis, material characterization, and real-time process monitoring. These sensors provide valuable insights into molecular composition and structural properties, enabling the precise identification and quantification of chemical species in diverse environments [1,2].

In practical applications, spectroscopic measurements are influenced by various physical factors. Changes in sample properties, such as viscosity and temperature, as well as variations in the measurement environment, for instance optical path length, can introduce nonlinearities that distort the spectral response through both additive and multiplicative effects. These distortions may compromise measurement accuracy and reliability. Thus, it is essential to use advanced algorithms to compensate for these unwanted influences and ensure robust data interpretation. In this contribution, we have only considered temperature variations, as this is a common perturbation in the process industry (e.g., in fermentation tanks) and is also easy to recreate for testing. However, the approach can be easily extended to account for other types of perturbations.

Several authors have proposed data-driven approaches for modeling spectral responses without assuming predefined functional dependencies on wavelength. In the context of this work, an overview by [3] examines data-driven methods designed to mitigate the impact of temperature variations on Near-Infrared (NIR) spectra. Notable algorithms include Extended Multiplicative Signal Correction [4], Local and Global Partial Least-Squares calibration strategies [5], Extended Loading Space Standardization, and Systematic Error Prediction [6]. While these data-driven approaches have demonstrated effectiveness, they rely on an informative dataset for calibration, which is typically performed offline. In addition, they lack the flexibility to integrate prior knowledge, limiting their adaptability in dynamic environments.

To overcome these limitations, we propose modeling the spectral response using physics-based spectral methods [7,8]. This approach represents the spectral response through parametric models, formulated as a linear combination of peak functions that carry physicochemical significance. These models inherently account for nonlinear effects, such as peak shifts and deformations, providing a more interpretable and adaptable framework for spectral analysis.

Following the physics-based approach, the integration of the sensor model with the process dynamic model leads to a nonlinear state-space representation, where the output equation is nonlinear and contains unknown parameters. Although Kalman filters are standard algorithms for state estimation in linear time-varying systems, the model in this application does not conform to a standard structure. However, it can be reformulated in a standard form by applying a transformation as suggested in [9] for a continuous-time model. This transformation results in a time-variant linear output map, where the previously unknown parameters appear linearly in the state equation multiplying the inputs, facilitating their estimation within a structured framework. For this structure, ref. [10] presents an approach based on two interconnected Kalman-like filters, and ref. [11] proposes one based on the combination of a Kalman filter and recursive least square (RLS) algorithm. Kalman filters for linear time-varying systems have been shown to exhibit superior convergence speed and capacity to tune the filter dynamic compared to adaptive observers, such as those described in [12].

The convenient structure of the model obtained through state transformation comes at the cost of introducing additional state variables, resulting in an overparameterized model and increased computational effort for estimation. To address this issue, an alternative error prediction approach has been proposed in [13], which avoids the need for state transformation but leads to highly complex recursive equations for computing the sensitivity of the prediction error with respect to states and parameters.

To address the aforementioned limitations, the present contribution extends the author’s earlier work in continuous time [14] by incorporating a discrete-time formulation, more sophisticated and realistic sensor models, and the explicit consideration of process disturbances. Moreover, it builds upon the results presented in [15] by conducting a formal observability analysis and introducing a novel reduced-order estimator designed to decrease the dimensionality of the system and, consequently, reduce computational complexity. In addition, a thorough comparative evaluation of the proposed estimators is provided, benchmarked against the standard Extended Kalman Filter.

The paper is organized as follows: Section 2 describes an integrated discrete-time model for the dynamic aspects of the process and the spectroscopic information. Section 3 presents a standard Extended Kalman Filter and two versions of an Adaptive Kalman Filter for performing online calibration. It also provides a convergence analysis of the AKF, considering the specific structure of the model to shed light on the conditions that need to be fulfilled. Section 4 presents an example that addresses the problem of correcting the effect of an external variable on the mixture of two components. It compares the performance of all filters and performs a sensitivity analysis with respect to external signal, noise level, and initial values for parameters and states. Finally, the last section summarizes the conclusions and outlines future work.

## 2. Process and Sensor Modeling

Consider a process in which the concentration of each component in the mixture is independently controlled and monitored by a spectroscopic sensor, as shown in Figure 1.

The transport and mixing dynamic is represented by a discrete-time linear model as follows:(1)$$\begin{array}{ccc} x(k+1) & = & Ax\left(k\right)+Bu\left(k\right)+\omega_{x}\left(k\right) \end{array}$$
$$\begin{array}{ccc} w(k) & = & Cx(k) \end{array}$$
where $u\left(k\right)={\left[u_{1}\left(k\right),\cdots ,u_{m}\left(k\right)\right]}^{T}$ is the input vector, $x={\left[x_{1}\left(k\right),\cdots ,x_{n}\left(k\right)\right]}^{T}$ is the state vector, and $w\left(k\right)={\left[w_{1}\left(k\right),\cdots ,w_{m}\left(k\right)\right]}^{T}$ denotes the concentration vector. The state disturbance vector, $\omega \left(k\right)={\left[\omega_{1}\left(k\right),\cdots ,\omega_{m}\left(k\right)\right]}^{T}$, follows an independent Gaussian noise process with zero mean and a covariance matrix $\Sigma_{\omega_{x}}=\sigma_{\omega_{x}}I$. The dimensions of the state-space model matrices are $A\in R^{n\times n}$, $B\in R^{n\times m}$, and $C\in R^{m\times n}$. The model assumes that A is Hurwitz and has been validated in previous research [16].

The sensor output, denoted as y(k,λ)∈R, represents the absorbance of the mixture and can be written, as described in [15], as(2)$$\begin{array}{c} y\left(k,\lambda \right)=\sum_{q=1}^{N_{p}}\theta_{q}\Psi_{q}\left(\lambda ,v\left(k\right)\right)w\left(k\right)+\nu \left(k,\lambda \right) \end{array}$$
where $\Psi_{q}\left(\lambda ,v\left(k\right)\right)$ contains the cross products between the functions modeling the effect of the external signals and the peak function representing the spectral response of the sensor. The vector $w\left(k\right)={\left[w_{1}\left(k\right),\cdots ,w_{m}\left(k\right)\right]}^{T}$ is the concentration vector and θ is the vector of unknown calibration factors. The model also considers that the absorbance of different components is influenced by an exogenous variable v(k). We assume that the output disturbances ν(k,λ) are independent Gaussian noise processes with zero mean and covariance matrix $\Sigma_{\nu }\left(\lambda \right)=\sigma_{\nu }\left(\lambda \right)$. Notice that Equation (Equation 2) can also be written as(3)$$\begin{array}{c} y\left(k,\lambda \right)=\theta^{T}\Psi \left(\lambda ,v\left(k\right)\right)w\left(k\right)+\nu \left(k,\lambda \right) \end{array}$$
where $\theta^{T}=\left[\theta_{1},\cdots ,\theta_{Np}\right]$ and $\Psi \left(\lambda ,v\left(k\right)\right)={\left[\Psi_{1}\left(\lambda ,v\left(k\right)\right),\cdots ,\Psi_{Np}\left(\lambda ,v\left(k\right)\right)\right]}^{T}$.

**Remark** **1.**
*While the sensor model is primarily designed for spectroscopic sensors, its framework can also accommodate sensors in other applications where sensor characteristics depend on external factors and may vary over time. Drifting sensor gains and nonlinear behaviors are common in various tactical sensors. Several Kalman filters have been proposed in this field (see, for instance, [17,18]). The method proposed in this paper could potentially be applied to these contexts, although further research is required.*


## 3. On-Line Correction

The on-line correction methods estimate concentrations based on spectral measurements, even though they are affected by exogenous variables. Thus, the on-line calibration problem can be defined as follows:

Given the process and sensor models, Equations (Equation 1) and (Equation 2), and the measurements of u(k), y(k,λ) and v(k), respectively, the problem is to estimate the concentrations w(k) and the calibration factors $\theta_{i}$.

We consider the unknown parameters to be constants perturbed by a zero mean stochastic variable, satisfying the following equation,(4)$$\theta_{i}\left(k+1\right)=\theta_{i}\left(k\right)+\omega_{\theta_{i}}\left(k\right),\;\;\;i=1,\cdots N_{p}\;$$

### 3.1. Extended Kalman Filter

By augmenting the state vector with the parameters, i.e., $x_{e}={\left[x\;\;\theta \right]}^{T}$, Equations (Equation 1), (Equation 2) and (Equation 4) can be written as follows:(5)$$x_{e}\left(k+1\right)=A_{e}x_{e}\left(k\right)+B_{e}u\left(k\right)+\omega_{x_{e}}\left(k\right)$$
where(6)$$A_{e}=\left[\begin{array}{cc} A & 0\\ 0 & I \end{array}\right]\;\;B_{e}=\left[\begin{array}{c} B\\ 0 \end{array}\right]$$
and $\omega_{x_{e}}\left(k\right)={\left[\omega_{x}\left(k\right)\;\;\omega_{\theta }\left(k\right)\right]}^{T}$.

The recursive equations for the EKF update equation is(7)$${\hat{x}}_{e}\left(k\right)=A_{e}{\hat{x}}_{e}\left(k-1\right)+B_{e}u\left(k-1\right)+K\left(k,\lambda \right)e\left(k,\lambda \right)$$
where the estimation error is(8)$$\begin{array}{cc} e(k,\lambda )= & y\left(k,\lambda \right)-\hat{\theta }{\left(k\right)}^{T}\Psi \left(\lambda ,v\left(k\right)\right)C\hat{x}\left(k\right) \end{array}$$
for all λ∈Λ. The gain matrix K(k,λ) associated to the state is updated as follows:(9)$$\begin{array}{cc} P(k|k-1) & =A_{e}P\left(k-1|k-1\right)A_{e}^{T}+Q_{e} \end{array}$$(10)$$\begin{array}{cc} \Sigma (k,\lambda ) & =C_{e}\left(k,\lambda \right)P\left(k|k-1\right)C_{e}{\left(k,\lambda \right)}^{T}+R \end{array}$$(11)$$\begin{array}{cc} K(k,\lambda ) & =P\left(k|k-1\right)C_{e}{\left(k,\lambda \right)}^{T}\Sigma {\left(k,\lambda \right)}^{-1} \end{array}$$(12)$$\begin{array}{cc} P(k|k) & =[I-K\left(k,\lambda \right)C_{e}\left(k,\lambda \right)]P\left(k|k-1\right) \end{array}$$
where the linearized measurement model is given by $C_{e}\left(k,\lambda \right)=\left[C_{x}\left(k,\lambda \right)\;C_{\theta }\left(k,\lambda \right)\right]$ with(13)$$\begin{array}{cc} C_{x}\left(k,\lambda \right) & =\hat{\theta }{\left(k\right)}^{T}\Psi \left(\lambda ,v\left(k\right)\right)C \end{array}$$(14)$$\begin{array}{cc} C_{\theta }\left(k,\lambda \right) & =\Psi \left(\lambda ,v\left(k\right)\right)C\hat{x}\left(k\right) \end{array}$$
for all λ∈Λ. We note that the wavelength range Λ will in practice be discretized, and measurements at different wavelengths should either be stacked into a measurement vector, or the Kalman filter can be updated with each wavelength sequentially. The same applies to the AKF and its variant in the following sections.

**Remark** **2.**
*In a stochastic framework, the matrices $Q_{e}$ and R represent noise covariance values, capturing the uncertainty associated with system dynamics and measurements. Conversely, in a deterministic approach, where randomness is not considered, these matrices function as tuning parameters. This perspective allows for adjusting $Q_{e}$ and R according to specific needs or prior knowledge, treating them as hyperparameters or regularization terms in the estimation process.*


**Remark** **3.**
*The accuracy of estimated values in the Extended Kalman Filter (EKF) depends on several key factors, such as the precision of linearization, the observability of the system, the nature of noise, initial state conditions, and the underlying system dynamics. Since these elements play a crucial role in the effectiveness of EKF implementation, the specific criteria for achieving convergence will vary based on the particular application.*


### 3.2. Adaptive Kalman Filter

To transform the nonlinear model into a time-variant linear model, it is necessary to define a new variable $q_{i}\left(k\right)=\theta_{i}\left(k\right)x\left(k\right)$. This approach was proposed by [9] for continuous-time. By multiplying the recursive equation of (Equation 1) by $\theta_{i}\left(k\right)$ we derive the following over-parameterized time-variant linear model:(15)$$\begin{array}{ccc} q_{i}\left(k+1\right) & = & Aq_{i}\left(k\right)+\theta_{i}\left(k\right)Bu\left(k\right)+\theta_{i}\left(k\right)\omega_{x}\left(k\right)+\\ & & x\left(k+1\right)\omega_{\theta_{i}}\left(k\right)\\ y(k,\lambda ) & = & \sum_{i=1}^{N_{p}}\Psi_{i}\left(\lambda ,v\left(k\right)\right)Cq_{i}\left(k\right)+\nu \left(k,\lambda \right) \end{array}$$

A drawback of this overparameterization is that it neglects certain dependencies between redundant variables. As a result, the initially simple noise properties become more intricate and exhibit correlations. Consequently, noise is not explicitly taken into account in the following equations.

By defining $p={\left[x\;\;q_{1}\cdots q_{N_{p}}\right]}^{T}$ and $\theta ={\left[\theta_{1}\;\cdots \theta_{N_{p}}\right]}^{T}$, the system Equations (Equation 4) and (Equation 15) can be written as(16)$$\begin{array}{ccc} \left[\begin{array}{c} p(k+1)\\ \theta (k+1) \end{array}\right] & = & \left[\begin{array}{cc} A_{p} & A_{\theta }\left(k\right)\\ 0 & I \end{array}\right]\left[\begin{array}{c} p(k)\\ \theta (k) \end{array}\right]+\left[\begin{array}{c} B_{p}\\ 0 \end{array}\right]u\left(k\right)\\ y(k,\lambda ) & = & C_{p}\left(k,\lambda \right)p\left(k\right) \end{array}$$
where(17)$$A_{p}=\left[\begin{array}{cccc} A & 0 & 0 & 0\\ 0 & A & 0 & 0\\ 0 & 0 & \ddots & 0\\ 0 & 0 & 0 & A \end{array}\right]B_{p}=\left[\begin{array}{c} B\\ 0\\ \vdots \\ 0 \end{array}\right]$$(18)$$A_{\theta }\left(k\right)=\left[\begin{array}{ccc} 0 & 0 & 0\\ Bu(k) & 0 & 0\\ 0 & \ddots & 0\\ 0 & 0 & Bu(k) \end{array}\right]$$(19)$$C_{p}\left(k,\lambda \right)=\left[\begin{array}{cccc} 0 & \Psi_{1}\left(\lambda ,v\left(k\right)\right)C & \cdots & \Psi_{N_{p}}\left(\lambda ,v\left(k\right)\right)C \end{array}\right]$$

An Adaptive Kalman Filter with distinct error dynamics for the state and parameters can be derived using the algorithm proposed by Zhang [11], resulting in the following equations: $$\begin{array}{cc} \hat{p}\left(k\right) & =A_{p}\hat{p}\left(k-1\right)+A_{\theta }\left(k\right)\hat{\theta }\left(k-1\right)+B_{p}u\left(k-1\right)+ \end{array}$$(20)$$\begin{array}{cc} \; & K\left(k,\lambda \right)e\left(k,\lambda \right)+\Upsilon \left(k\right)[\hat{\theta }\left(k\right)-\hat{\theta }\left(k-1\right)] \end{array}$$(21)$$\begin{array}{cc} \hat{\theta }\left(k\right) & =\hat{\theta }\left(k-1\right)+\Gamma \left(k,\lambda \right)e\left(k,\lambda \right) \end{array}$$
where the estimation error is(22)$$\begin{array}{cc} e(k,\lambda )= & y\left(k,\lambda \right)-C_{p}\left(k,\lambda \right)\left[A_{p}\hat{p}\left(k-1\right)+A_{\theta }\left(k\right)\hat{\theta }\left(k-1\right)+B_{p}u\left(k\right)\right]. \end{array}$$

The state gain matrix K(k,λ) is updated as follows: (23)$$\begin{array}{cc} K(k,\lambda ) & =P\left(k|k-1\right)C_{p}{\left(k,\lambda \right)}^{T}\Sigma {\left(k,\lambda \right)}^{-1} \end{array}$$(24)$$\begin{array}{cc} P(k|k-1) & =A_{p}P\left(k-1|k-1\right)A_{p}^{T}+Q \end{array}$$(25)$$\begin{array}{cc} \Sigma (k,\lambda ) & =C_{p}\left(k,\lambda \right)P\left(k|k-1\right)C_{p}{\left(k,\lambda \right)}^{T}+R \end{array}$$(26)$$\begin{array}{cc} P(k|k) & =[I-K\left(k,\lambda \right)C_{p}\left(k,\lambda \right)]P\left(k|k-1\right) \end{array}$$

**Remark** **4.**
*Equations (24)–(26) calculate the matrices involved in determining the state gain matrix, denoted as K(k,λ). The matrices Q and R are tuning parameters. Selecting their values can be challenging because, due to the model’s overparameterization, they no longer represent the noise covariances.*


The gain matrix Γ(k,λ) associated with the parameters update is given by the following recursive equations: (27)$$\begin{array}{cc} \Gamma (k,\lambda ) & =S\left(k-1\right)\Omega {\left(k,\lambda \right)}^{T}\Lambda \left(k,\lambda \right) \end{array}$$(28)$$\begin{array}{cc} \Omega (k,\lambda ) & =C_{p}\left(k,\lambda \right)A_{p}\Upsilon \left(k-1\right)+C_{p}\left(k,\lambda \right)A_{\theta }\left(k\right) \end{array}$$(29)$$\begin{array}{cc} \Lambda (k,\lambda ) & ={\left[\gamma \Sigma \left(k,\lambda \right)+\Omega \left(k\right)S\left(k-1\right)\Omega {\left(k\right)}^{T}\right]}^{-1} \end{array}$$(30)$$\begin{array}{cc} S(k) & =\frac{1}{\gamma }S\left(k-1\right)-\frac{1}{\gamma }S\left(k-1\right)\Omega {\left(k,\lambda \right)}^{T}\Lambda \left(k,\lambda \right)\Omega \left(k,\lambda \right)S\left(k-1\right) \end{array}$$
where γ∈(0,1) is a forgetting factor and Υ(k) is an auxiliary variable defined as follows:(31)$$\begin{array}{cc} \Upsilon (k) & =\left[I-K\left(k,\lambda \right)C_{p}\left(k,\lambda \right)\right]A_{p}\Upsilon \left(k-1\right)+\left[I-K\left(k,\lambda \right)C_{p}\left(k,\lambda \right)\right]A_{\theta }\left(k\right). \end{array}$$

**Remark** **5.**
*Equations (Equation 28)–(Equation 31) calculate matrices associated with the parameter gain matrix, Γ(k,λ), and the gain matrix related to the estimated parameter variations, Υ(k), as utilized in the state estimation equation. These equations can be interpreted as a recursive least square estimator with a forgetting factor [11].*


### 3.3. Reduced Order Adaptive Kalman Filter

A reduced order AKF (rAKF) can be obtained if the original states x are not considered in the extended model, i.e., p just contains a function of the new state variables q, $p={\left[q_{1}\cdots q_{N_{p}}\right]}^{T}$. Hence, the system equations can be written as(32)$$\begin{array}{ccc} p(k+1) & = & A_{p}p\left(k\right)+B_{\theta }\left(k\right)\theta \left(k\right) \end{array}$$
$$\begin{array}{ccc} y(k,\lambda ) & = & C_{p}\left(k,\lambda \right)p\left(k\right) \end{array}$$
where(33)$$A_{p}=\left[\begin{array}{cccc} A & 0 & 0 & 0\\ 0 & A & 0 & 0\\ 0 & 0 & \ddots & 0\\ 0 & 0 & 0 & A \end{array}\right]$$(34)$$B_{\theta }\left(k\right)=\left[\begin{array}{ccc} Bu(k) & 0 & 0\\ 0 & \ddots & 0\\ 0 & 0 & Bu(k) \end{array}\right]$$(35)$$C_{p}\left(k,\lambda \right)=\left[\begin{array}{ccc} \Psi_{1}\left(\lambda ,v\left(k\right)\right)C & \cdots & \Psi_{N_{p}}\left(\lambda ,v\left(k\right)\right)C \end{array}\right].$$

The estimation of q and θ is the obtained by using Equations (Equation 20) and (Equation 21). The original state estimate can then be derived by solving the following least-squares problem:(36)$$\min_{x}{∥\mathrm{diag}\left(\hat{\theta }\left(k\right)\right)x-{\left[{\hat{q}}_{1}\left(k\right)\cdots {\hat{q}}_{N_{p}}\left(k\right)\right]}^{T}∥}^{2}.$$

Thus,(37)$$\hat{x}\left(k\right)=\frac{\mathrm{diag}(\hat{\theta }\left(k\right))}{\hat{\theta }{\left(k\right)}^{T}\hat{\theta }\left(k\right)}{\left[{\hat{q}}_{1}\left(k\right)\cdots {\hat{q}}_{N_{p}}\left(k\right)\right]}^{T}$$

### 3.4. Convergence Properties

The Adaptive Kalman Filters will converge if a set of key assumptions associated to the model structure and persistency of excitation for the input signals, i.e., u(k) and v(k), are satisfied [11].

**Assumption** **A1.**
*The pairs $\left[A_{p},\;Q\right]$ and $\left[A_{p},\;C_{p}\left(k,\lambda \right)\right]$ are uniformly completely controllable and observable, respectively.*


Given that $A_{p}$ is Hurwitz, it is always possible to choose a positive definite matrix Q so that the pair $\left[A_{p},\;Q\right]$ is uniformly completely controllable, i.e., there are exit constants $c_{c}\in R$ and $N_{c}\in N$ such that the controllability Grammian satisfies the following:(38)$$W_{c}\left(k+N_{c},k\right)\geq c_{c}I\;\;\forall k\geq 0$$

On other hand, if v(k) satisfies the condition that there are exit constants $c_{o}\in R$ and $N_{o}\in N$ such that observability Grammian satisfies the following:(39)$$W_{o}\left(k+N_{o},k\right)\geq c_{o}{A_{p}^{N_{0}}}^{T}A_{p}^{N_{0}}\;\;\forall k\geq 0$$
then the pair $\left[A_{p},\;C_{p}\left(k,\lambda \right)\right]$ will be uniformly completely controllable. This condition is crucial for ensuring the parameter convergence, and it implies that the exogenous signal must exhibit a form of persistent excitation.

**Remark** **6.**
*Assumption 1 only considers the conditions to ensure the boundedness of P(k|k), Σ(k,λ), and K(k,λ).*


**Remark** **7.***The interpretation of $C_{p}\left(k,\lambda \right)$ in* Assumption 1 *is as a matrix of stacked vectors, each representing the values corresponding to a wavelength belonging to the set* Λ.

**Remark** **8.**
*Notice that for the full state model, in Equation (Equation 17), in the case of the AKF, the first elements of $C_{p}\left(k,\lambda \right)$ corresponding to the state vector, x, are zeros, and the matrix $A_{p}$ has a block-diagonal structure. This structure renders the state variable x unobservable, as the observability Gramian will have a block of zeros on the diagonal. In contrast, the equations for the rAKF, Equation (*
Equation 32
*), lead to an observability Gramian with non-zero blocks on its diagonal.*


**Assumption** **A2.**
*The input signal u(k) is persistently exciting in the sense that there exists an integer h>1 and a constant α∈R such that ∀k≥0*

(40)
$$\sum_{l=0}^{h-1}\Omega^{T}\left(k+l,\lambda \right)\Sigma {\left(k+l,\lambda \right)}^{-1}\Omega \left(k+l,\lambda \right)\geq \alpha I_{p}$$

*where Σ(k,λ) and Ω(k,λ) are defined by Equations (25) and (28), respectively.*


Assumption 2 deals with the evolution of $A_{\theta }$ in (Equation 32) and imposes conditions on the input signal, u(k). This assumption is required to ensure the boundedness of S(k). Notice that Ω(k,λ) is related to the input signal through $A_{\theta }$ in Equations (Equation 28) and (Equation 31).

**Remark** **9.**
*The Assumptions 1 and 2 require that the input and external variable satisfy persistent excitation condition, which in practice is not always satisfied. Recent works have addressed this problem [19,20], and the analysis of these approaches in the context of this application is the subject of future work.*


## 4. Experimental and Simulation Results

This section presents both experimental and simulation-based results to illustrate and evaluate the proposed modeling and state estimation approach. First, the modeling methodology is demonstrated using experimental data. Following this, a simulated case study is introduced to explore the behavior and performance of the proposed state estimation algorithm under a broad range of operating conditions, highlighting its robustness and effectiveness.

To illustrate the modeling approach using experimental data, we consider the water absorbance spectra in the wavelength range of 850 nm to 1050 nm at different temperatures, as shown in Figure 2. The dataset used for this purpose comes from [5].

Following a physics-based approach, the water spectra ϕ(λ,T) can be represented as a linear combination of peak functions $\psi_{i}\left(\lambda \right)$ as follows:(41)$$\phi \left(\lambda ,T\right)=\sum_{i=1}^{5}a_{i}\left(T\right)\psi_{i}\left(\lambda \right)=\sum_{i=1}^{5}a_{i}\left(T\right)\exp\left(-\frac{{\left(\lambda -c_{i}\right)}^{2}}{w_{i}^{2}}\right)$$
where the peak functions are defined in terms of Gaussian parameters, namely the center $c_{i}$ and width $w_{i}$. A two-step approach is used to identify the model parameters and validate this structure against experimental data.

First, for a given temperature, all parameters are estimated by solving a nonlinear least-squares problem fitted to the spectral response. The resulting basis functions, scaled by their respective linear coefficients, are shown in Figure 3.

In the second step, the Gaussian parameters ($c_{i}$, $w_{i}$) are kept fixed, and the linear coefficients $a_{i}\left(T\right)$ are computed for all temperature conditions. These coefficients are then regressed against temperature. The results, depicted in Figure 4, confirm the validity of the assumed model structure—in this case, a linear dependence on temperature. The estimated parameters and corresponding regression results are summarized in Table 1. The maximum approximation error across the entire dataset is less than 1.

After having validated the modeling approach on the basis of real spectral responses, we illustrate the main ideas of the estimator by considering an example of the simulated mixing of two components presented in [15], where $u_{1}\left(k\right)$ and $u_{2}\left(k\right)$ are the flow rates of each component, v(k) is the mixture temperature, and y(k,λ) is the measured spectrum.

The discrete state space model, which represents the transport and the mixing of the two components, is defined by the following matrices:(42)$$A=\left[\begin{array}{cccc} 1.6 & -0.8 & 0 & 0\\ 0.8 & 0 & 0 & 0\\ 0 & 0 & 1.7 & -0.85\\ 0 & 0 & 0.85 & 0 \end{array}\right]B=\left[\begin{array}{cc} 1 & 0\\ 0 & 0\\ 0 & 1\\ 0 & 0 \end{array}\right]$$(43)$$C=\left[\begin{array}{cccc} 0 & 0.05 & 0 & 0\\ 0 & 0 & 0 & 0.0265 \end{array}\right]$$

The measured spectrum considers the absorbance spectra of each component and the effect of temperature, i.e.,(44)$$y\left(k,\lambda \right)=w_{1}\left(k\right)\phi_{1}\left(\lambda ,v\left(k\right)\right)+w_{2}\left(k\right)\phi_{2}\left(\lambda ,v\left(k\right)\right)$$
where absorbance of each component is modeled as a linear combination of two Gaussian peaks, as illustrated in Figure 5a, and described by the following parameterization:(45)$$\begin{array}{cc} \phi_{1}\left(\lambda ,v\left(k\right)\right) & =\theta_{1}\psi_{11}\left(\lambda \right)+\left(\theta_{2}+\theta_{3}v\left(k\right)\right)\psi_{12}\left(\lambda \right) \end{array}$$
$$\begin{array}{cc} \phi_{2}\left(\lambda ,v\left(k\right)\right) & =\left(\theta_{4}+\theta_{5}v\left(k\right)\right)\psi_{21}\left(\lambda \right)+\theta_{6}\psi_{22}\left(\lambda \right) \end{array}$$
where $\psi_{ij}\left(\lambda \right)$ represents Gaussian peaks associated to the model of each component *i*. Note that in this case, the external variable v(k) influences the absorbance of each component linearly, as illustrated in Figure 5b.

Rearranging terms (Equation 44) can be written as Equation (Equation 2), where the set of parameters $\left\{\theta_{1},\theta_{2},\theta_{3},\theta_{4},\theta_{5},\theta_{6}\right\}$ represents the parameters of the sensor model, which are assumed to be unknown. Their nominal values are $\theta ={\left[2.2,\;-1.5,\;0.15,\;-0.02,\;0.1,\;2.5\right]}^{T}$, and vectors $\Psi_{i}\left(\lambda ,v\left(k\right)\right)$ are as follows:(46)$$\begin{array}{c} \Psi_{1}={\left[\begin{array}{c} \psi_{11}\left(\lambda \right)\\ 0 \end{array}\right]}^{T},\;\Psi_{2}={\left[\begin{array}{c} \psi_{12}\left(\lambda \right)\\ 0 \end{array}\right]}^{T},\;\Psi_{3}={\left[\begin{array}{c} \psi_{12}\left(\lambda \right)v\left(k\right)\\ 0 \end{array}\right]}^{T} \end{array}$$
$$\begin{array}{c} \Psi_{4}={\left[\begin{array}{c} 0\\ \psi_{21}\left(\lambda \right) \end{array}\right]}^{T},\;\Psi_{5}={\left[\begin{array}{c} 0\\ \psi_{21}\left(\lambda \right)v\left(k\right) \end{array}\right]}^{T},\;\Psi_{6}={\left[\begin{array}{c} 0\\ \psi_{22}\left(\lambda \right) \end{array}\right]}^{T} \end{array}$$

The simulation examples initially focus on a base case to highlight the core features of the algorithms under specific conditions. Subsequently, a sensitivity analysis explores the variations in initial parameters, state conditions, and noise levels to further distinguish the algorithms’ performance under different scenarios.

### 4.1. Base Case

The base case considers zero-mean Gaussian disturbances in both the sensor measurements and the state equations, with variances $\sigma_{\nu }=0.5$ and $\sigma_{\omega }=0.1$, respectively. It also accounts for step changes in the flow rates of each component and periodic variations in the external variable, as shown in Figure 6. The effect of these changes and the noise on the spectral response can be seen in Figure 7.

In the next simulations for both the EKF and AKF, the initial conditions for the estimated parameters are set to zero and the states are set as ${\left[100,\;100,\;50,\;50\right]}^{T}$. To illustrate the evolution of estimated states and parameters, we consider one realization of state and measurement noise.

For the EKF, we consider the following parameters and initial conditions: the matrix Q=0.1I is chosen to be a diagonal matrix, and $P\left(0\right)=10^{3}I$. As illustrated in Figure 8a, the estimated parameters converge to their true values more rapidly than the state. This observation is further supported by the evolution of the state and parameter error norms shown in Figure 8b.

For the AKF, the matrix Q=0.0001I was chosen as a constant diagonal matrix, $P\left(0\right)=10^{3}I$, and λ=1. The observer initial conditions for q are set to zero. The estimated concentrations quickly converge to the real values, although some parameters take longer to stabilize, as shown in Figure 9a. Variability in input variables, such as flow rates and temperature, ensures the convergence of both estimated concentrations and parameters. Furthermore, the state error asymptotically converges to zero, as depicted in Figure 9b.

In adaptive applications, once the algorithm has converged, it is always informative to analyze the structure of the final observer gain; in this case, it is K(300,λ). From Figure 10, two important observations can be drawn: first, the gains associated with the estimation of the state variables x are zero; second, the wavelength dependencies are linked to the location of the basis functions (see Figure 5a). Zero gains indicate a lack of observability for the associated states, which is consistent with Remark 8.

The evolution of the estimated parameters for the rAKF follows a similar pattern to that of the AKF, as shown in Figure 11a. However, as illustrated in Figure 11a,b, the estimated concentrations exhibit much faster dynamics, driven by the observer design rather than the open-loop system dynamics.

A comparative analysis of the dynamic responses, depicted in Figure 8, Figure 9 and Figure 11, shows that both the AKF and rAKF algorithms are effective in parameter estimation. The rAKF provides faster dynamics for estimated concentrations because of its enhanced observer design. However, the EKF takes longer to converge, with the state convergence being much slower than the parameter convergence.

A comprehensive analysis across multiple realizations enhances the robustness and reliability of the results by capturing the variability introduced by different instances of noise. Table 2 summarizes the results for one hundred noise realizations, presenting the mean values and standard deviations of the Root Mean Squared Errors (RMSEs) for both states and parameters. The results indicate that EKF exhibits higher mean values of $RMS_{x}$ and $RMS_{\theta }$ compared to the AKF and rAKF. Lower RMSE values generally suggest better performance, indicating that the AKF and rAKF outperform the EKF based on these measures. Although the rAKF initially shows larger errors than the AKF, its convergence is faster. For detailed information about the accuracy of each estimate, refer to the boxplot of the estimated parameters in Figure 12. It is evident that both the AKF and rAKF provide more accurate parameter estimates compared to the EKF.

### 4.2. Sensitivity Analysis

The following simulations demonstrate the sensitivity of the estimates to variations in the external signal, initial parameters, states, and noise levels. The values of Q, R, and P(0) are consistent with those used in the base case.

To analyze the effect of the external signal profiles, two additional profiles are considered. The first profile consists of a constant value of 20 (°C) with step changes occurring at time instant 75, 150, and 225. This profile provides less dynamic information than the one considered in the base case. The second profile is simply a constant value of 20 (°C), which does not provide enough information for estimating the parameters, violating one of the assumptions necessary for convergence.

Based on the results summarized in Table 3, the comparison of estimation algorithms—the EKF, AKF, and rAKF—under two distinct signal profiles, Steps and Constant, reveals distinct performance characteristics. The Adaptive Kalman Filter (AKF) consistently demonstrates superior performance in both signal scenarios, showing the lowest mean Root Mean Square (RMS) error for the state x and θ, coupled with the least variability. This indicates that the AKF provides more accurate and reliable estimates across different conditions. In contrast, the Extended Kalman Filter (EKF) exhibits the highest mean RMS errors and greater variability for both state variables, making it less consistent and accurate compared to the AKF. The reduced order Adaptive Kalman Filter (rAKF) performs better than the EKF but falls short of the AKF, presenting intermediate levels of mean errors and variability. Specifically, while the rAKF matches the AKF’s performance in estimating θ under both signal profiles, it does not achieve the same accuracy for the states. A comparative analysis with respect to the results of the more dynamic rich input signal (Table 2) shows that the EKF performs poorly in estimating the state variables and parameters. However, the AKF and rAKF provide similar estimates for the state variables. As expected, all algorithms perform worse in terms of parameter estimates when the signal has fewer time variations. The lack of persistent excitation in the AKF and rAKF mainly affects the parameter estimation.

To analyze the effect of initial conditions, two sets are considered: Initial Condition 1, where the state and parameters are zero, and Initial Condition 2, where $\hat{x}\left(0\right)={\left[3,\;6,\;9,\;8\right]}^{T}$ and $\hat{\theta }\left(0\right)={\left[1,\;1,\;0.1,\;0.1,\;0.1,\;1\right]}^{T}$. The results summarized in Table 4 clearly show that the EKF is highly sensitive to the choice of initial conditions, whereas the AKF and rAKF provide consistent results regardless of initial conditions. From a practical perspective, this robustness in initialization is a significant advantage, simplifying the setup of the estimation algorithms.

Finally, Table 5 summarizes the results of the algorithms with respect to an increase of 50% and 100% of the noise level considered in the base case. At the 50% noise level, the AKF demonstrates the lowest mean RMS error for the parameters with the smallest variability, indicating superior robustness to moderate noise. The rAKF follows closely, performing slightly better than the EKF in terms of both parameters and state estimation accuracy. The EKF, while reasonably effective, shows higher mean errors and variability.

At the higher noise level of 100%, all algorithms exhibit increased mean RMS errors and variability. The AKF maintains a relatively better performance, although its state error increases compared to the 50% noise scenario. The rAKF shows a performance comparable to the AKF, with a slight increase in errors and variability, but it still outperforms the EKF. The EKF displays the highest mean RMS errors and variability in both state variables under both noise levels. Overall, while all algorithms are affected by increased noise, the AKF and rAKF exhibit better robustness and consistency in estimation performance compared to the EKF.

## 5. Conclusions

This research introduces a novel discrete-time correction algorithm based on physic-based models and an Adaptive Kalman Filter (AKF) for spectroscopic applications. The AKF is formulated using an extended model of the process and the spectroscopic sensor, enabling the simultaneous estimation of system states and calibration factors while accounting for the influence of external variables. Additionally, an observability analysis of the AKF led to the development of a reduced-order AKF (rAKF), designed to improve computational efficiency by reducing the number of estimated variables.

The results demonstrate that the AKF provides robust and accurate state estimation across various conditions, outperforming both the rAKF and the Extended Kalman Filter (EKF). The rAKF offers a computationally efficient alternative with faster convergence, making it suitable for scenarios where reduced complexity is necessary. In contrast, the EKF exhibits higher estimation errors and variability, confirming the advantages of the proposed approaches over conventional filtering techniques.

These findings highlight the potential of the AKF and rAKF for enhancing spectroscopic sensing, particularly in applications such as Near-Infrared (NIR) and Raman spectroscopy. However, the practical implementation of these filters requires careful consideration of the persistent excitation conditions necessary for ensuring asymptotic convergence. Since real-world applications may not always satisfy these conditions, future research should focus on developing relaxed convergence criteria to ensure stable performance even under limited excitation.

The promising performance of the proposed AKF framework suggests significant opportunities for advancing spectroscopic sensing technologies. Future studies will explore real-time laboratory validations and extend the methodology to broader sensing applications, further refining the capabilities of adaptive filtering in spectroscopic analysis.

## Figures and Tables

**Figure 1 sensors-25-02513-f001:**
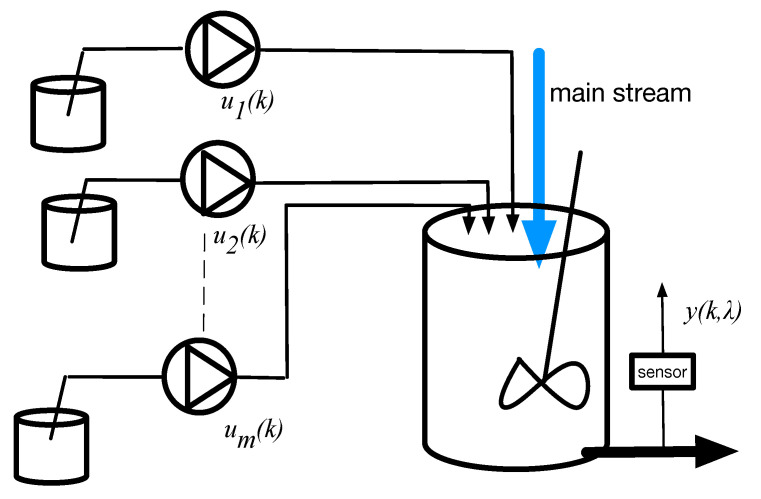
Basic mixing process.

**Figure 2 sensors-25-02513-f002:**
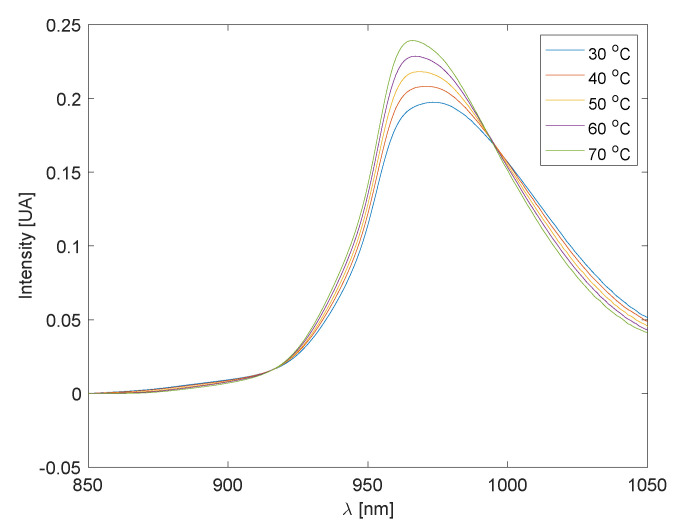
Water spectra for different temperatures.

**Figure 3 sensors-25-02513-f003:**
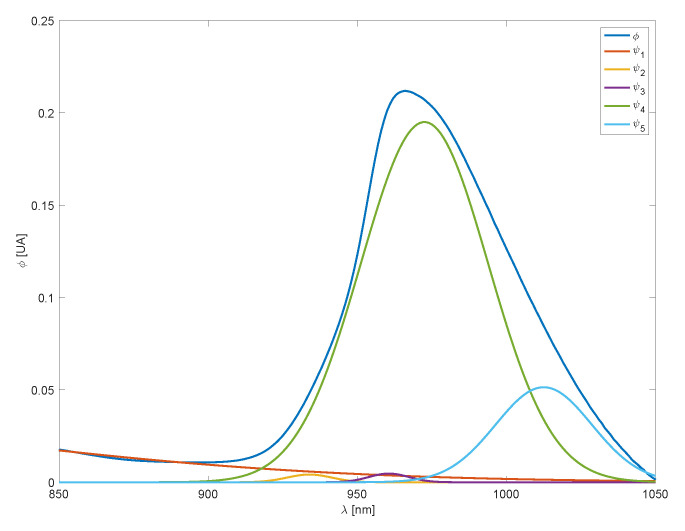
Water spectra at 60 °C and basis functions.

**Figure 4 sensors-25-02513-f004:**
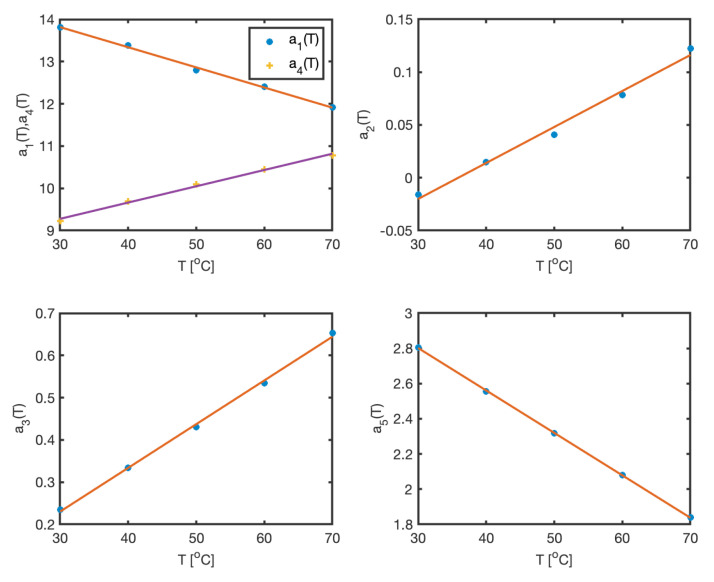
Linear dependence of coefficients on temperature.

**Figure 5 sensors-25-02513-f005:**
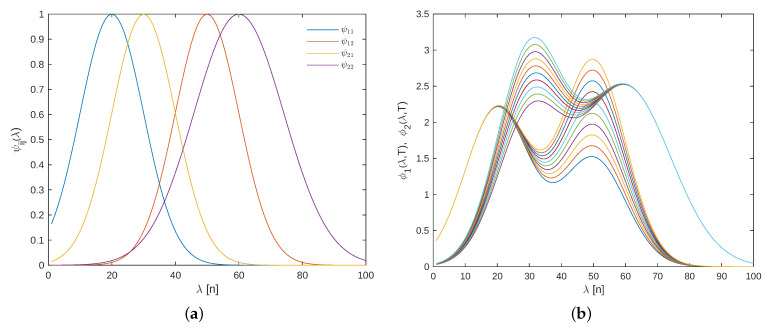
(**a**) Basis spectral functions. (**b**) Spectral response of each component for different temperatures.

**Figure 6 sensors-25-02513-f006:**
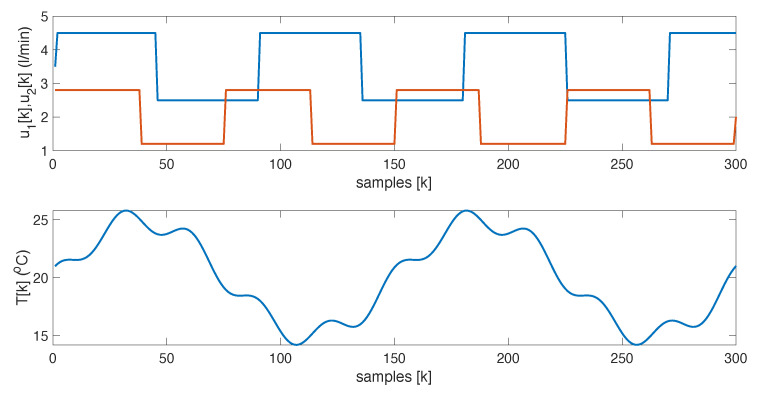
Flow rates and mixture temperature.

**Figure 7 sensors-25-02513-f007:**
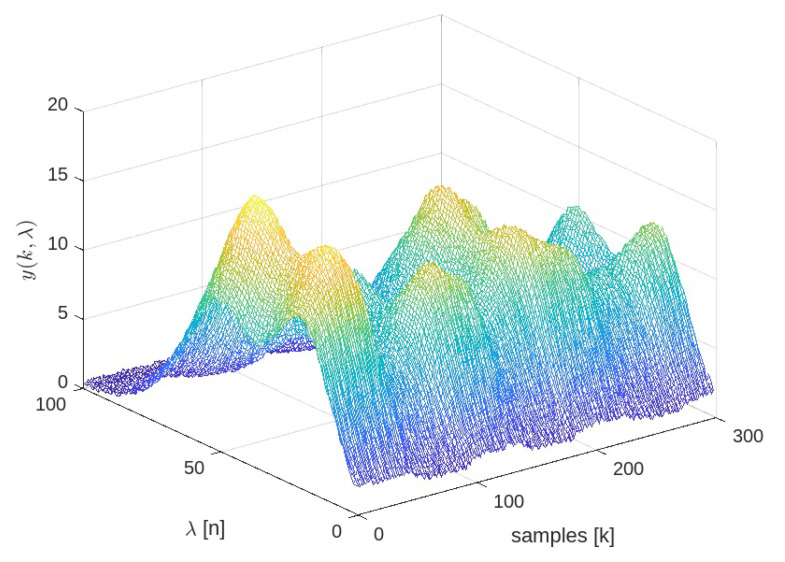
Spectral-time sensor response. The color represents the intensity of the spectral signal.

**Figure 8 sensors-25-02513-f008:**
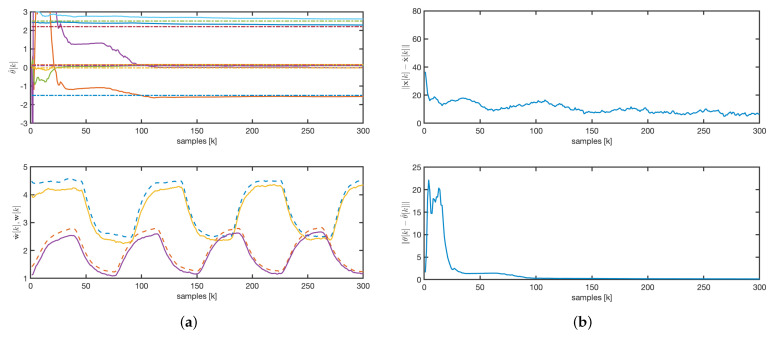
EKF: (**a**) parameters and concentrations evolution. Solid line: estimated values. Dashed line: real values. (**b**) State and parameters error norm.

**Figure 9 sensors-25-02513-f009:**
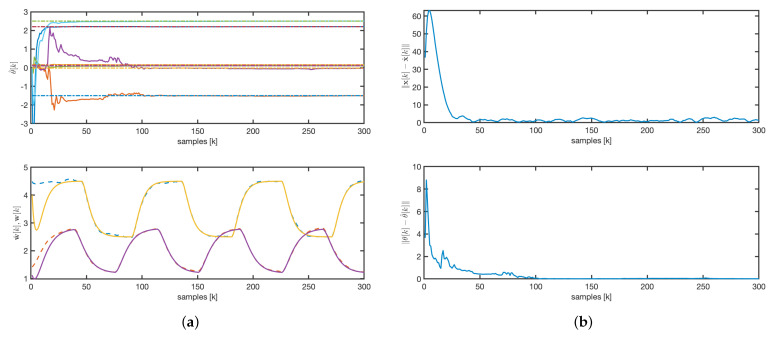
AKF: (**a**) parameters and concentrations evolution. Solid line: estimated values. Dashed line: real values. (**b**) State and parameters error norm.

**Figure 10 sensors-25-02513-f010:**
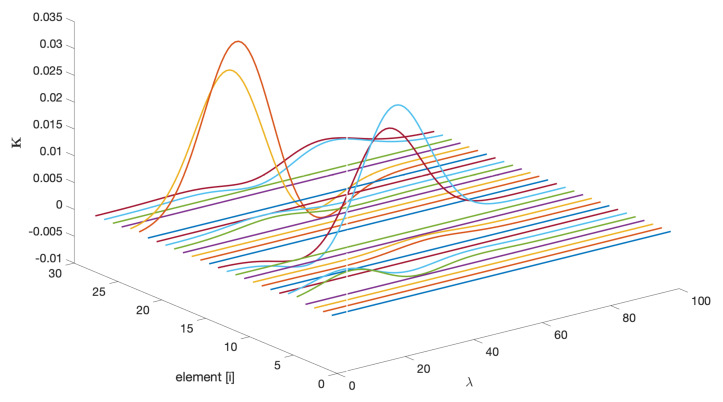
Elements of the AKF final vector gain as a function of λ.

**Figure 11 sensors-25-02513-f011:**
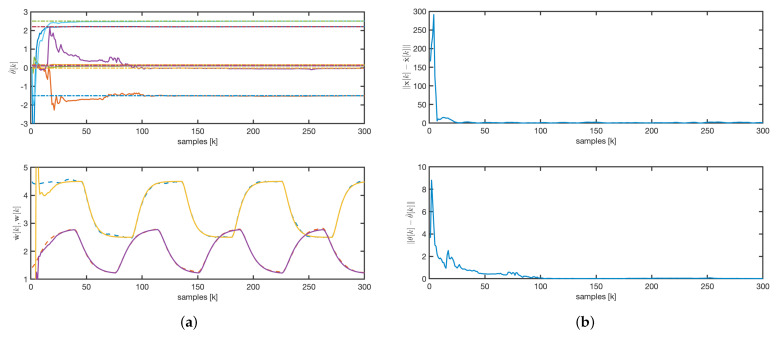
rAKF: (**a**) parameters and concentrations evolution. Solid line: estimated values. Dashed line: real values. (**b**) State and parameters error norm.

**Figure 12 sensors-25-02513-f012:**
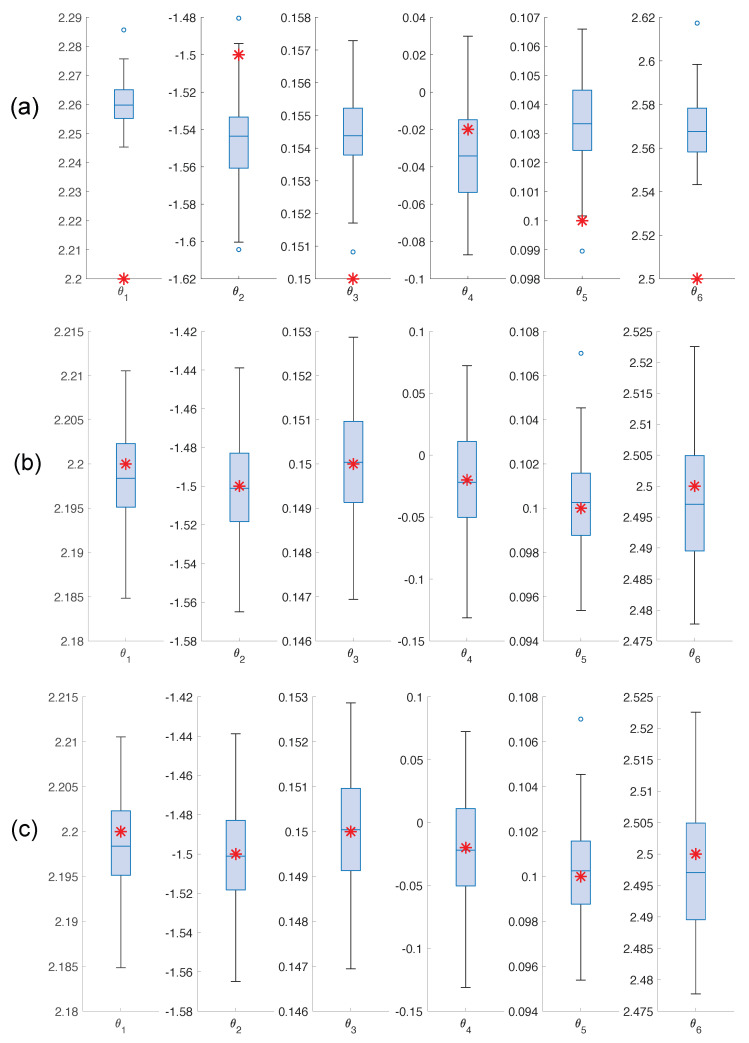
Boxplot for the parameter estimates (**a**) EKF (**b**) AKF (**c**) rAKF. A red asterisk represents the values of the real parameters.

**Table 1 sensors-25-02513-t001:** Parameters for water absorbance spectra model.

*i*	*a_i_*(*T*)	*c_i_*	*w_i_*
1	15.2452−0.04765T	688.8	294.58
2	−0.10288+0.00308T	9333.9	17.7178
3	−0.0801+0.01035T	960.6	15.68581
4	8.1216+0.03854T	972.5	50.31
5	3.5253+0.03411T	960.6	37.97104

**Table 2 sensors-25-02513-t002:** Performance measures.

Algorithm	$\mu_{{\mathrm{RMS}}_{x}}$	$\sigma_{{\mathrm{RMS}}_{x}}$	$\mu_{{\mathrm{RMS}}_{\theta }}$	$\sigma_{{\mathrm{RMS}}_{\theta }}$
EKF	2.6827	0.1486	0.8822	0.0778
AKF	2.0344	0.0468	0.5514	0.0376
rAKF	2.4546	0.0888	0.5514	0.0376

**Table 3 sensors-25-02513-t003:** Performance measures for different external signal profiles.

Signal	Algorithm	$\mu_{{\mathrm{RMS}}_{x}}$	$\sigma_{{\mathrm{RMS}}_{x}}$	$\mu_{{\mathrm{RMS}}_{\theta }}$	$\sigma_{{\mathrm{RMS}}_{\theta }}$
Steps	EKF	3.3590	0.2089	0.7815	0.0233
	AKF	2.0344	0.0468	0.7179	0.0164
	rAKF	2.4344	0.0864	0.7179	0.0164
Constant	EKF	3.3955	0.2128	1.2370	0.0030
	AKF	2.0344	0.0468	1.2593	0.0022
	rAKF	2.4342	0.0864	1.2593	0.0022

**Table 4 sensors-25-02513-t004:** Performance measures for different initial conditions.

IC	Algorithm	$\mu_{{\mathrm{RMS}}_{x}}$	$\sigma_{{\mathrm{RMS}}_{x}}$	$\mu_{{\mathrm{RMS}}_{\theta }}$	$\sigma_{{\mathrm{RMS}}_{\theta }}$
IC 1	EKF	11.6040	0.2106	10.1255	0.5335
	AKF	2.5395	0.0377	0.5514	0.0376
	rAKF	2.2728	0.0620	0.5724	0.0364
IC 2	EKF	6.7165	0.1064	6.0134	0.1754
	AKF	2.5678	0.0373	0.5488	0.0377
	rAKF	2.2728	0.0620	0.5724	0.0364

**Table 5 sensors-25-02513-t005:** Performance measures for different noise level.

Noise Level	Algorithm	$\mu_{{\mathrm{RMS}}_{x}}$	$\sigma_{{\mathrm{RMS}}_{x}}$	$\mu_{{\mathrm{RMS}}_{\theta }}$	$\sigma_{{\mathrm{RMS}}_{\theta }}$
50%	EKF	2.7151	0.2052	0.9501	0.1058
	AKF	2.7035	0.0468	0.5514	0.0376
	rAKF	2.5971	0.1008	0.6104	0.0481
100%	EKF	2.7621	0.2569	1.0244	0.1304
	AKF	2.8338	0.0667	0.6680	0.0577
	rAKF	2.7341	0.1041	0.6700	0.0575

## Data Availability

The water NIR spectra is available from https://github.com/salvadorgarciamunoz/eiot/tree/master/pyEIOT, accessed on 8 April 2025. Please note that these data were produced by Wülfert et al. [5], and all copyright remains with the authors.

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
