# Peer review of "Adaptive Kalman Filtering for Compensating External Effects in On-Line Spectroscopic Measurements"

_sensors, 2025, doi:10.3390/s25082513_

Round 1
Reviewer 1 Report
Comments and Suggestions for Authors
The manuscript is remarkably interesting and well formed. But there are some notes that should be clarified.
- It will be more convincing if supplement real-world spectroscopic experimental data (e.g., from laboratory or industrial scenarios) to validate the algorithm’s performance under actual disturbances.
- Please include a comparison with the latest adaptive filtering methods (e.g., the combination of deep reinforcement learning and Kalman filtering) to highlight the competitiveness of the Adaptive Kalman Filter (AKF).
- The differences between the core ideas of the AKF (such as the state variable transformation) and existing methods should be more clearly articulated to better highlight the originality.
Author Response
Comment 1: It will be more convincing if supplement real-world spectroscopic experimental data (e.g., from laboratory or industrial scenarios) to validate the algorithm’s performance under actual disturbances.
Response 1: We have expanded the Introduction to provide additional context on a practical real-world application where the proposed method can be effectively deployed. Moreover, we have included an illustrative example using real spectroscopic data to demonstrate the effectiveness of the modeling approach. This example, based on authentic absorbance spectra, highlights the method's ability to capture the key features of temperature-dependent spectral behavior and confirms its compatibility with widely used techniques such as near-infrared (NIR) spectroscopy.
While full experimental validation of the estimation algorithm under real industrial conditions is planned for future work, the simulation studies presented here provide meaningful insights into the algorithm's robustness and practical applicability. Specifically, the simulations consider varying initial conditions, different temporal profiles of external perturbations (e.g., temperature drift and fluctuations), and multiple levels of measurement and process noise. These scenarios were designed to closely mimic real-world operating conditions and to rigorously evaluate the performance of the proposed method under realistic challenges.
Comment 2: Please include a comparison with the latest adaptive filtering methods (e.g., the combination of deep reinforcement learning and Kalman filtering) to highlight the competitiveness of the Adaptive Kalman Filter (AKF).
Response 2: Thank you for the suggestion. The primary focus of this work is to develop a state estimation strategy grounded in a physics-based modeling approach. This results in a parsimonious and interpretable model, which stands in contrast to data-driven methods such as those based on deep learning or reinforcement learning. While recent approaches that combine deep reinforcement learning with Kalman filtering are promising, they typically require large amounts of training data, high computational resources, and may lack interpretability—factors that are critical in many industrial applications. To evaluate the performance of the proposed algorithm, we compare it against the standard Extended Kalman Filter (EKF), which remains a widely used benchmark for nonlinear systems. A comprehensive comparison with data-driven adaptive filtering methods is certainly of interest and is considered a valuable direction for future work.
Comment 3: The differences between the core ideas of the AKF (such as the state variable transformation) and existing methods should be more clearly articulated to better highlight the originality.
Response 3: We have added a paragraph to the Introduction that clarifies the core ideas behind the state transformation used in the Adaptive Kalman Filter (AKF). Specifically, we explain how the transformation yields a model with a time-varying linear output map, where the unknown parameters appear linearly in the state equations through their interaction with the inputs. This structure facilitates parameter estimation within a well-defined and interpretable framework. However, this advantage comes at the cost of introducing additional state variables, resulting in an overparameterized model and increased computational demand. To address this limitation, we have included references to our previous work and clearly highlighted how the proposed approach contributes a novel solution namely, a reduced-order formulation that preserves structure while improving computational efficiency. These changes can be found in page 2 lines 69-82.
Reviewer 2 Report
Comments and Suggestions for Authors
The manuscript entitled “Adaptive Kalman filtering for compensating external effects in on-line spectroscopic measurements” is focused on developing advanced spectroscopic sensing technologies by enabling the simultaneous estimation of system states and calibration factors while accounting for the influence of external variables.
My overall view about the manuscript is positive, as it is well written and clearly presented in all sections.
The paper would be recommended for this journal after the authors address the following issue:
What is the novelty of this method when compared to their prior work? In the introduction, it is mentioned that this research extends their previous work in continuous time [13], but this reference is from 2020, and the group already published similar results in 2023 and 2024.
Author Response
Comment 1: What is the novelty of this method when compared to their prior work? In the introduction, it is mentioned that this research extends their previous work in continuous time [13], but this reference is from 2020, and the group already published similar results in 2023 and 2024.
Response 1: While this work builds on our earlier contributions, it introduces several key advancements. The continuous-time formulation in [14] is extended here to a discrete-time setting, enabling practical implementation. Compared to [15], which relies on a full state transformation, we propose a novel reduced-order estimator that significantly reduces computational complexity by limiting the number of state variables.
In [13] an alternative error-prediction method was introduced to avoid state transformation, but it involves complex recursive sensitivity equations. The current work offers a more computationally efficient solution while maintaining interpretability and structure. Additionally, we incorporate realistic sensor models, explicitly consider disturbances, and perform a formal observability analysis—none of which were fully addressed in the previous studies. A comparative evaluation with the Extended Kalman Filter further demonstrates the practical relevance of the proposed approach. These changes can be found in page 2 lines 69-82.
Reviewer 3 Report
Comments and Suggestions for Authors
The authors propose adaptive Kalman filtering for the processing of spectral data, which is not a novel concept.
The authors claim in the abstract that “This study addresses the challenges of real-time spectroscopic sensing in industrial applications, where external factors such as temperature fluctuations, pressure variations, and particle size distribution significantly impact measurement accuracy”. In the simulations only temperature is considered.
To increase the interest of the paper for the Sensors community, the performance of the proposed algorithm should be tested on a more realistic scenario. If no real spectral data is available, at least some detail of the process this can be applied to should be given, and also it is relevant the spectroscopic technique used in the experiment.
Author Response
Comment 1: The authors propose adaptive Kalman filtering for the processing of spectral data, which is not a novel concept.
Response 1: We agree that Adaptive Kalman Filtering (AKF) is not a novel concept; however, the novelty of this work lies in its specific application to simultaneously estimate concentrations while compensating for external perturbations—particularly temperature variations—using a physically-informed model. This dual objective introduces unique challenges that are not typically addressed in standard AKF applications. As a result, the proposed formulation includes tailored modifications to the standard AKF framework, including model structure, reduced-order estimation, and observability considerations, which are specifically designed to address the characteristics of this problem. To the best of our knowledge, this integrated approach has not been previously reported in the literature.
Comment 2: The authors claim in the abstract that “This study addresses the challenges of real-time spectroscopic sensing in industrial applications, where external factors such as temperature fluctuations, pressure variations, and particle size distribution significantly impact measurement accuracy”. In the simulations only temperature is considered.
Response 2: While the abstract refers broadly to external perturbations such as temperature, pressure, and particle size, this study focuses specifically on temperature variations. This choice is motivated by the fact that temperature is one of the most common and impactful disturbances in industrial spectroscopic applications (e.g., fermentation or chemical reactors) and is also relatively straightforward to reproduce in a controlled testing environment. We acknowledge the importance of other factors such as pressure and particle size, and we emphasize that the proposed framework is general and can be extended to accommodate these additional sources of variability. This clarification has been added to the Introduction to more accurately reflect the scope of the current study. See page 2 lines 35-38.
Comment 3: To increase the interest of the paper for the Sensors community, the performance of the proposed algorithm should be tested on a more realistic scenario.
Response 3: While the current study is based on simulations, we have conducted a comprehensive and systematic evaluation of the proposed algorithm under a range of realistic and practically relevant conditions. These include varying initial conditions, different temporal profiles of external perturbations (e.g., temperature drift and fluctuations), and multiple levels of measurement and process noise. These scenarios were carefully designed to reflect typical challenges encountered in real-world spectroscopic sensing applications. We believe this thorough testing provides valuable insight into the robustness and practical applicability of the method. Nonetheless, we acknowledge the importance of experimental validation and consider it a key direction for future work.
Comment 4: If no real spectral data is available, at least some detail of the process this can be applied to should be given, and also it is relevant the spectroscopic technique used in the experiment.
Response 4: We have added in the Introduction a brief description of an industrial process where the proposed approach is applicable, highlighting its relevance to real-time spectroscopic monitoring. Additionally, we have incorporated a modeling example based on experimental spectral data to demonstrate that the proposed framework can effectively approximate real absorbance spectra. This example illustrates the suitability of the method for practical scenarios and confirms its compatibility with spectroscopic techniques such as near-infrared (NIR) spectroscopy, which is commonly used in industrial applications. See page 8 lines 225-251.
Round 2
Reviewer 3 Report
Comments and Suggestions for Authors
My recommendation is to perform the simulation using a more realistic scenario as they have already started in the revised version of the manuscript. In this version they simulate how the water spectrum varies with temperatura based on the data of their reference [5], but the results of their methd are given with two non specific compounds. Their reference [5] also provides data to simulate ethanol and iso-propanol. They could use these compounds to perform the simulations of modifing the flow rate and temperature fluctuations and test the algorithm with the simulation of these specific compounds (water and ethanol for example, or others).
Author Response
Comment: My recommendation is to perform the simulation using a more realistic scenario as they have already started in the revised version of the manuscript. In this version they simulate how the water spectrum varies with temperatura based on the data of their reference [5], but the results of their methd are given with two non specific compounds. Their reference [5] also provides data to simulate ethanol and iso-propanol. They could use these compounds to perform the simulations of modifing the flow rate and temperature fluctuations and test the algorithm with the simulation of these specific compounds (water and ethanol for example, or others).
Response:
We appreciate your recommendation. In the present study, we modelled pure water as a representative system to demonstrate the applicability of the proposed modelling framework in capturing temperature-induced spectral variations. The decision to omit the dataset from [5], which involves mixtures of additional compounds, was intentional, as the primary objective of this work is to characterize the performance of various estimation algorithms and conduct a rigorous comparative analysis. Incorporating the dataset from [5] would require substantial contextualisation and methodological elaboration, thereby expanding the scope and length of the manuscript beyond its intended focus. A comprehensive treatment, including the integration of the most effective estimation algorithm identified in this study, will be presented in a forthcoming contribution specifically focused on modelling and dynamic compensation of external perturbations in solutions containing organic compounds.